# Structurally diverse macrocycle co-crystals for solid-state luminescence modulation

Bin Li[1], Lingling Liu[1], Yuan Wang[1], Kun Liu[1], Zhe Zheng[1], Shougang Sun[2], Yongxu Hu[2], Liqiang Li �[2] & Chunju Li ⓘ[1] ✉

Organic co-crystals offer an opportunity to fabricate organic functional materials. Traditional co-crystals are generally packed following the segregated or mixed stacking mode, leading to the lack of structural and functional diversity. Herein, we report three sets of macrocycle co-crystals with identical co-constitutions. The macrocycle co-crystals differ in the stoichiometric ratios (2:1, 1:1, and 2:3) of the constituents and molecular packing modes. The co-crystals are constructed using triangular pyrene-macrocycle and 1,2,4,5-tetra-cyanobenzene exploiting exo-wall charge-transfer interactions. Interestingly, the three co-crystals exhibit distinct, tunable emission properties. The corresponding emission peaks appear at 575, 602, and 635 nm, covering yellow via orange to red. The X-ray diffraction analyses and the density functional theory calculations reveal the superstructure-property relationships that is attributed to the formation of different ratios of charge-transfer transition states between the donor and acceptor motifs, resulting in red-shifted luminescence.

Organic co-crystal engineering involves assembling two or more types of chemical species in stoichiometric ratios into highly ordered superstructures[1]. The assemblies are formed by exploiting inter-molecular noncovalent interactions[2,3], and this strategy has been used in recent years to fabricate multifunctional materials that exhibit unpredicted and versatile chemico-physical properties[4–23]. Unlike tra-ditional chemical synthesis involving covalent bond breakage and formation, organic co-crystal engineering possesses the advantages for creating functional materials, including: i) the preparation techniques are facile and low-cost, and do not include the execution of compli-cated synthetic procedures and harsh experimental conditions; ii) the structures, sizes, morphologies, and stoichiometric ratios associated with the co-crystals can be tuned by selecting suitable conformers or changing solvents; iii) the co-crystals not only retain the inherent properties of the individual component but also exhibit multi-functional properties via the synthetic and synergistic effects exerted by the constituent units. To date, organic co-crystals have been extensively applied in organic semiconductor[6–8], ferroelectricity[9–11], fluorescence and organic room-temperature phosphorescence[12–15], stimuli-responsive materials[16–18], pharmaceutics[19,20], etc.

Organic co-crystals, built by planar small molecules, are most commonly packed following the segregated- and mixed-stacking modes. Macrocycles with intrinsic cavities and polygonal structures are principal tools of supramolecular chemistry[24–31]. In theory, macrocycle-based organic co-crystals should exhibit various mole-cular arrangements[32]. These co-crystals should be characterized by tunable stoichiometric ratios and interesting properties, the origin of which can be traced back to the polygonal topological structures and the presence of interior cavities. To date, few reports on macrocycle co-crystals (MCCs) based on naphthalenediimide-based molecular triangle[33,34], pillar[n]arene[35–38], and pagoda[4]arene[39] have been reported by Stoddart' group, Huang' group, Chen' group and us. It is challenging to construct MCCs with identical co-components but varying donor (D)-acceptor (A) stoichiometries and study the superstructure–property relationships characterizing the molecules.

Herein, we report three sets of MCCs constructed by a pyrene-macrocycle (Pe[3]) as the donor and a 1,2,4,5-tetracyanobenzene (TCNB) unit as the acceptor (Fig. 1). The MCCs were fabricated by exploiting exo-wall charge transfer (CT) interactions. The D-A stoi-chiometries were varied (2:1, 1:1, and 2:3), and the molecular

[1]Academy of Interdisciplinary Studies on Intelligent Molecules, Tianjin Key Laboratory of Structure and Performance for Functional Molecules, College of Chemistry, Tianjin Normal University, Tianjin 300387, PR China. [2]Tianjin Key Laboratory of Molecular Optoelectronic Sciences, Department of Chemistry, Institute of Molecular Aggregation Science, Tianjin University, Tianjin 300072, PR China. ✉e-mail: cjli@shu.edu.cn

**Fig. 1 | Chemical structures.** The components (Pe[3], Pe, and TCNB) of CT co-crystals.

arrangements were tuned. Intriguingly, these MCCs exhibited solid-state yellow, orange, and red luminescence properties. Results from crystal structure analyses and density functional theory (DTF) calculations revealed that the emission characteristics of the molecules were influenced by the D-A ratios associated with the CT co-crystals.

## Results

### Synthesis and crystal structure of Pe[3]

A triangular macrocycle Pe[3] bearing three pyrene units in the skeletal structure was selected as the co-crystal component as the pyrene group functions as an electron-rich donor and luminophore. Modular synthetic strategies previously reported by us[40,41] for the fabrication of macrocycles were followed to synthesize Pe[3] (Supplementary Figs. 1 and 2). In the solid-state structure, Pe[3] crystallizes in the trigonal R-3 space group. Two pairs of enantiomers were tightly stacked to form a supramolecular tetramer characterized by a one-dimensional tubular structure developed exploiting C–H⋯π, C–H⋯O, and π⋯π stacking interactions (Supplementary Figs. 3 and 4). The packing of the supramolecular tetramer in the *ab* plane accounted for the formation of a four-layer honeycomb 2D architecture. The molecular arrangements of each layer are presented in the Supplementary Fig. 5.

### Solid-state superstructures of MCCs

We attempted to develop macrocycle-based co-crystals by carefully regulating the crystallization conditions and using pyrene and TCNB units as the building blocks, as these were suitable conformers that could be efficiently used for the development of co-crystals exploiting CT interactions[42–44]. It was found that solvent modulation played a critical role in controlling the D-A stoichiometric ratio and molecular arrangement in the crystal superstructures. Co-crystallization of electron-rich Pe[3] with electron-deficient TCNB at a ratio of 1:3 in tetrahydrofuran (THF), dioxane, and CHCl₃ resulted in the formation of three sets of CT co-crystals, represented as MCC-1, MCC-2, and MCC-3. The Pe[3]:TCNB molar ratios in the crystal structures were 2:1, 1:1, and 2:3, respectively (Supplementary Figs. 6–8). The optical micrographs showed only one homogeneous crystals with specific morphology and color was formed for each MCC (Supplementary Fig. 9). ¹H NMR experiments of three MCCs were further given a strict stoichiometric ratio of MCC-1 for 2:1, MCC-2 for 1:1 and MCC-3 for 2:3, indicating the purity of MCCs (Supplementary Figs. 10–12). The crystallographic data corresponding to the three co-crystals are presented in the Supplementary Tables 1–3.

The MCCs crystallized in the triclinic P-1 space group. One Pe[3] and 0.5 TCNB molecules were associated with the asymmetric unit of MCC-1, two TCNB (occupancy factor: 0.5) were associated with MCC-2, and 1.5 TCNB were associated with MCC-3 (Supplementary Figs. 13–15). One pair of enantiomers of Pe[3] constituted the three MCCs

(Supplementary Fig. 16). It was noted that the conformations of Pe[3] in the four crystal structures (conformations of individual Pe[3] and three binary MCCs units) were different from each other. The methylene angle and the dihedral angle between the dimethoxybenzene and pyrene planes were different (Supplementary Fig. 17), proving that the macrocycle was flexible and adaptive.

The remarkable difference among the structures of the three co-crystals can be attributed to the CT participation ratio of macrocyclic skeleton (1/3 for MCC-1, 2/3 for MCC-2 and 3/3 for MCC-3) (Fig. 2). In MCC-1, one TCNB molecule binds with two Pe[3] molecules to form a sandwich-typed CT complex exploiting the face-to-face π⋯π interactions. The interplanar distance was calculated to be 3.28 Å (dihedral angle: 1.58°, Fig. 2a), and the molecule consisted of two C–H⋯O interactions. The distance between the hydrogen atoms in the TCNB units and the closest methoxy oxygen atoms in Pe[3] was calculated to be 2.73 Å (Supplementary Fig. 18). The Pe[3] molecule can interact with an adjacent Pe[3] unit to produce a parallelogram consisting of a supramolecular dimer between the pyrene units via the generation of parallel π⋯π stacking interactions. The average plane–plane distance was found to be 3.40 Å, and the corresponding dihedral angle was 0° (Supplementary Fig. 19). In the case of MCC-2, the two edges of Pe[3] interact with two types of crystallographically distinct TCNB molecules (TCNB-1 and TCNB-2) in a face-to-face fashion by CT interactions (Fig. 2b). Multiple C–H⋯N interactions between the nitrogen atoms of the cyano groups in the TCNB molecule and the closest methoxy and pyrenyl hydrogen atoms in Pe[3] further stabilized the CT complex (Supplementary Fig. 20). Each Pe[3] molecule interacted with neighboring Pe[3] molecules via C–H⋯O, C–H⋯π, and π⋯π interactions, forming a tetrameric basic unit (Supplementary Fig. 21). In the crystal structure of MCC-3, the three edges of the Pe[3] unit participated in generating CT interactions with two types of crystallographically distinct TCNB molecules (TCNB-3 and TCNB-4). The units interacted face-to-face to form a butterfly-shaped structure (Fig. 2c). Multiple C–H⋯N and C–H⋯O interactions between Pe[3] and TCNB stabilized the CT complex (Supplementary Figs. 22 and 23). The propagation of these non-covalent interactions (in the MCC-1, MCC-2, and MCC-3 molecules) in a 2D plane resulted in the formation of three different 2D superstructure networks (Fig. 2e–g).

### Solid-state superstructure of Pe-TCNB

To illustrate that the structural diversity of MCCs can be attributed to the topology of the macrocycles, we grew co-crystals of the monomer (Pe, Fig. 1) with TCNB in the above mentioned solvents (THF, dioxane and CHCl₃) under the same crystallization conditions. However, after many attempts, only binary CT co-crystals of Pe-TCNB were obtained in dioxane and individual Pe[3] crystallized out in THF and CHCl₃ (Supplementary Figs. 24–26 and Supplementary Tables 4–6). The

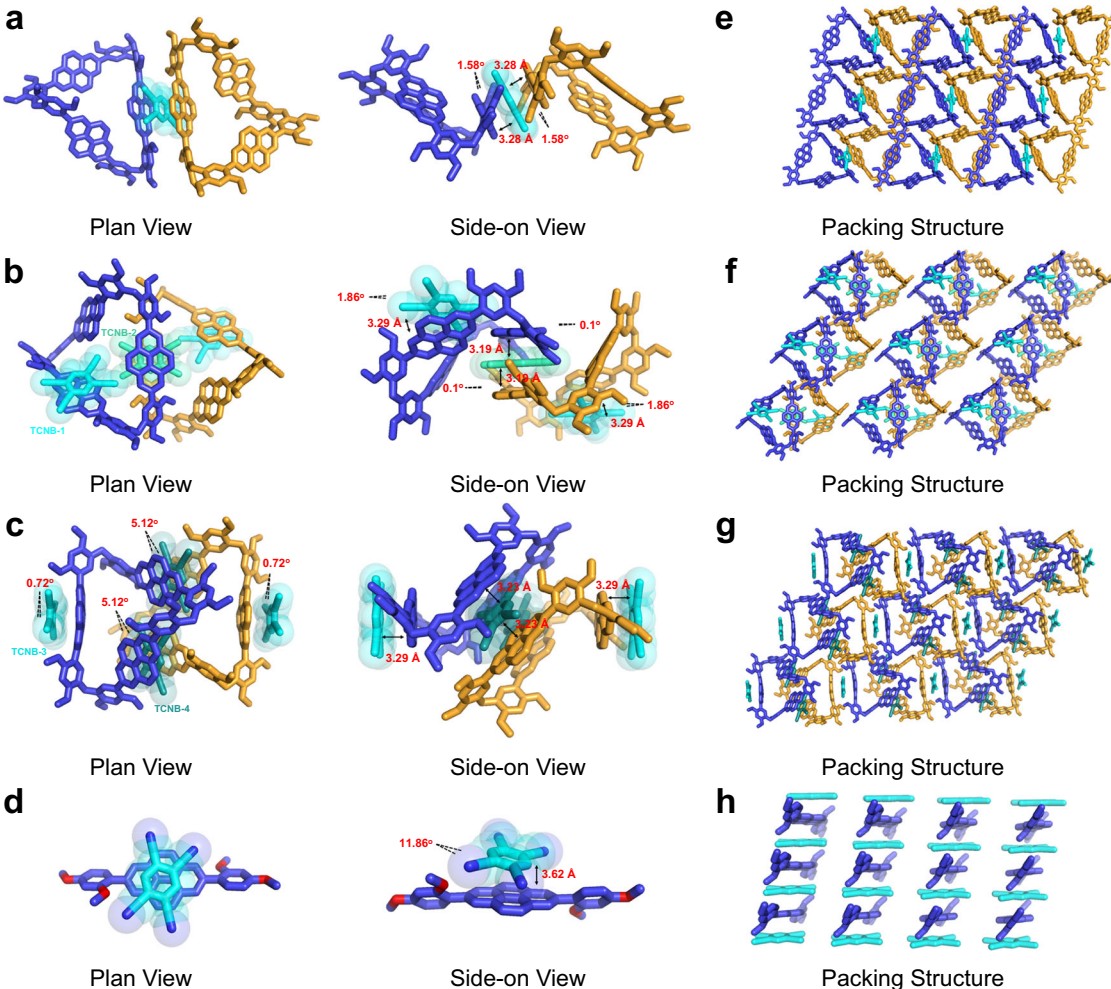

**Fig. 2 | Solid-state superstructures of MCC-1, MCC-2, MCC-3 and Pe-TCNB.**
**a–c** Schematic representation of the charge-transfer interactions and stoichiometric ratios between Pe[3] and TCNB in MCCs. **d** Crystal structure of Pe-TCNB. **e–h** Stacking modes of three MCCs and Pe-TCNB co-crystals in a 2D plane. Hydrogen atoms and solvents are omitted for clarity. Different colors represent symmetry equivalence. Crystallographically distinct TCNB molecules were marked as TCNB-1, TCNB-2, TCNB-3, and TCNB-4. The arrows represent π···π distances (Å) and the dashed lines represent dihedral angles.

Pe:TCNB ratio in the Pe-TCNB system was 1:1, and this was similar to the case of the traditional small-molecule CT co-crystals. The Pe and TCNB molecules were arranged alternately to form a column-like mixed stack (D–A–D–A–D) (Supplementary Fig. 25). The stacks were formed through the exploitation of π···π and C–H···N interactions (Fig. 2d). The columnar structure was extended via the formation of C–H···O interactions to form a 2D assembly (Fig. 2h and Supplementary Fig. 25). The results indicate that the polygonal skeleton of the macrocycles dictates the formation of structurally diverse MCCs.

### Photophysical properties of MCCs

Properties of CT-induced supramolecular assemblies have been demonstrated to modulate the optical characteristics of molecules[45]. Therefore, we explored the luminescence properties of the MCCs. The fluorescence microscopy images revealed that the three as-prepared co-crystals exhibited tunable luminescence properties. MCC-1 exhibited yellow luminescence, MCC-2 exhibited orange luminescence, and MCC-3 exhibited red luminescence (Fig. 3a). Individual Pe[3] crystals exhibited blue luminescence, and the results indicated that cocrystallization significantly affected the optical properties of the molecules. The luminescence behaviors of organic co-crystals are primarily influenced by the properties of either one donor and several acceptors or one acceptor and several donors that participate in CT interactions[46,47]. It has been rarely reported that organic co-crystals

characterized by identical supramolecular co-constitutions and varying D-A stoichiometries exhibit different luminescence behavior[48].

Solid-state ultraviolet-visible (UV-vis) absorption and fluorescence spectra were recorded and analyzed to elucidate the photophysical properties of the molecules before and after the formation of MCCs (Supplementary Figs. 27–32). As shown in the Supplementary Figs. 27–29, the three co-crystals displayed strong absorption bands, and the peaks corresponding to absorption appeared at 465 (for MCC-1), 497 (for MCC-2), and 532 nm (for MCC-3). The peaks were significantly red-shifted (by 159, 191, and 226 nm, respectively) compared to the peak corresponding to the individual components of Pe[3] crystals. The red shift could be attributed to the intermolecular CT interactions between Pe[3] and TCNB. The differences in the degrees of red-shift observed for the three CT co-crystals can be attributed to the more efficient D-A [π···π] overlaps in the crystal superstructures of MCC-3 than in that of MCC-2 and MCC-1.

Analysis of the fluorescence emission spectral profiles revealed bathochromic shifts of the peaks appearing in the profiles recorded for the three MCCs (Fig. 3b). Emission peaks appeared at 575, 602, and 635 nm, and the peak corresponding to the individual Pe[3] crystals appeared at 364 nm (Supplementary Figs. 30–32). The results agreed well with the results obtained by analyzing the fluorescence microscopy images (Fig. 3a). The International Commission on Illumination (CIE) coordinates of the Pe[3] crystal, and the MCC-1, MCC-2, and MCC-

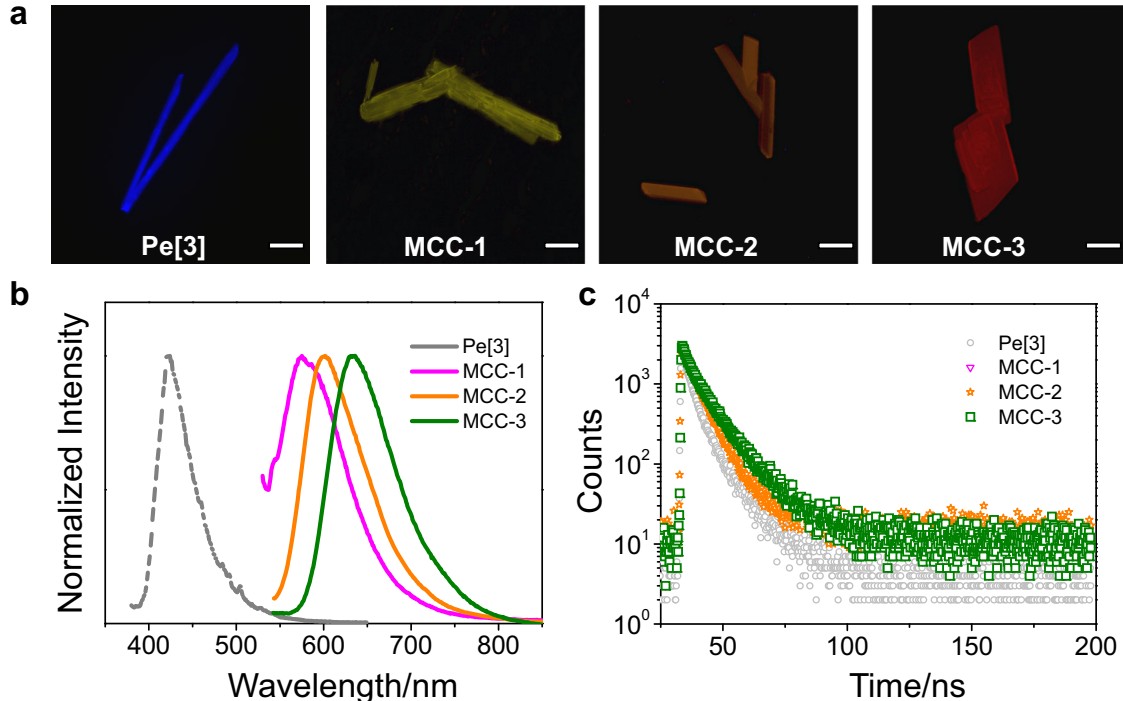

**Fig. 3 | Luminescence properties. a** Fluorescence microscopy images recorded for Pe[3], MCC-1, MCC-2, and MCC-3. Scale bar: 100 μm. The luminescence property could be tuned, and the MCCs exhibited yellow, orange, and red luminescence, respectively. **b** Solid-state fluorescence spectral profiles recorded for Pe[3] crystals and three MCCs. The peaks significantly red-shifted relative to the peak corresponding to the Pe[3] crystals. **c** Fluorescence decay curves corresponding to Pe[3] crystals and three MCCs.

3 co-crystals were found to be (0.16, 0.06), (0.51, 0.49), (0.54, 0.46), and (0.63, 0.37), respectively (Supplementary Fig. 33), based on the fluorescence spectral profiles. The results revealed that the luminescence color of the co-crystals could be tuned. Meanwhile, the solid-state photoluminescence quantum yield (PLQY, $\Phi_F$) and fluorescence lifetime ($\tau$) were also determined to better understand the optical properties (Supplementary Figs. 34–41). The results are listed in Table 1, and the corresponding decay curves are presented in Fig. 3c. The solid-state PLQYs of MCC-1, MCC-2, and MCC-3 were 1.2, 2.7, and 1.0%, respectively. The average lifetimes of the three co-crystals increased from MCC-1 (7.38 ns) to MCC-2 (9.38 ns) and MCC-3 (12.66 ns), reflecting the gradual enhancement of the degree of CT realized between the donor and acceptor units[46,47,49]. The results are consistent with the variations of the UV-vis absorption profiles.

**Frontier molecular orbitals calculations**
DFT calculations were performed using the hybrid B3LYP/6–31G functional to obtain deeper insights into the interplay between the structure and property of the MCCs. The structures required for the calculations were obtained by the single-crystal X-ray diffraction data. As shown in Fig. 4, the molecular orbital (MO) diagrams suggested that the highest occupied molecular orbitals (HOMOs) of MCC-1, MCC-2, and MCC-3 were concentrated on the electron-rich pyrene moiety. The energy levels were −5.18, −5.04, and −4.69 eV, respectively. These values were comparable to the values recorded for Pe[3] (−4.79 eV). The lowest

unoccupied molecular orbitals (LUMOs) of the three co-crystals were distributed on the electron-deficient TCNB molecule, and the approximate energy levels were −2.67, −2.74, and −2.71 eV for MCC-1, MCC-2, and MCC-3, respectively. The results indicated that CT interactions occurred when the charge was transferred from the HOMOs of Pe[3] to the LUMOs of TCNB. The gradual decrease in energy gaps (from 2.51 eV to 2.30 eV and subsequently to 1.98 eV) demonstrated an increased CT interaction. The tendency is in agreement with the results obtained from UV-Vis spectral profiles. The decrease in bandgap can be attributed to the change in the D-A ratio of the MCCs. The natural transition orbitals (NTO) analysis was carried out to analyze the excitation process of CT co-crystals (Supplementary Figs. 42 and 43).The results confirmed that the stoichiometric ratios dictated the structure of the molecules and the photophysical characteristics of the MCCs.

**Effect of solvents on MCCs formation**
To investigate the effect of solvents on the structure and stoichiometry of MCCs, we made many attempts and obtained 6 sets of MCCs and 2 individual macrocycle crystals (Supplementary Tables 8–15) from solvents with different morphologies and colors (Fig. 5a–h). Their crystal structures show that the formed D-A ratios of MCCs are 2:1 (in ClCH₂CH₂Cl and 1,3-dioxolane), 1:1 (in CH₂Cl₂, benzene and 2,3-dihydrofuran), 2:3 (in o-xylene) and 1:0 (in DMSO and DMF) (Supplementary Fig. 44). The varied stoichiometries of MCCs are depended on the solubility of TCNB (Fig. 5i). The lower the solubility, the stronger the solvophobic forces[50], and the CT participation ratio of TCNB is higher. Low solubility (<10 mM) of TCNB forms 2:3 MCC. Moderate solubility (41–85 mM) of TCNB tends to generate 1:1 MCC. High solubility (277–436 mM) of TCNB prefer to obtain 2:1 MCC. While TCNB solubility is more than 1000 mM, only individual macrocycles of Pe[3] were crystallized. The solubility (2.3–10 mM) of Pe[3] did not change too much in these solvents. Therefore, the effect of solvents on the diverse structures and tunable stoichiometric ratios of MCCs is decided by the

**Table 1 | Luminescent properties of the three MCCs**

| Co-crystal | $\lambda_{em}$ (nm) | $\Phi_F$ (%) | $\tau_{av}$ (ns) |
|---|---|---|---|
| MCC-1 | 575 | 1.2 | 7.38 |
| MCC-2 | 602 | 2.7 | 9.38 |
| MCC-3 | 635 | 1.0 | 12.66 |

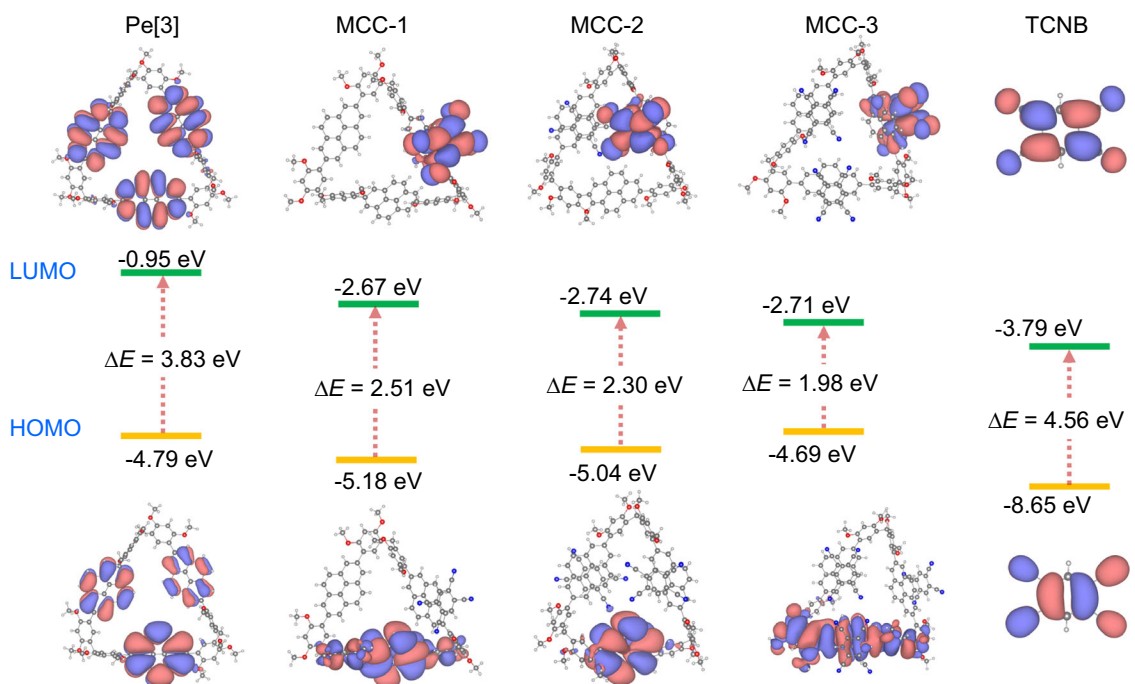

**Fig. 4 | DFT calculations.** Frontier molecular orbitals of Pe[3], MCC-1, MCC-2, MCC-3, and TCNB and the corresponding firstenergy gaps (3.83, 2.51, 2.30, 1.98, and 4.56 eV, respectively).

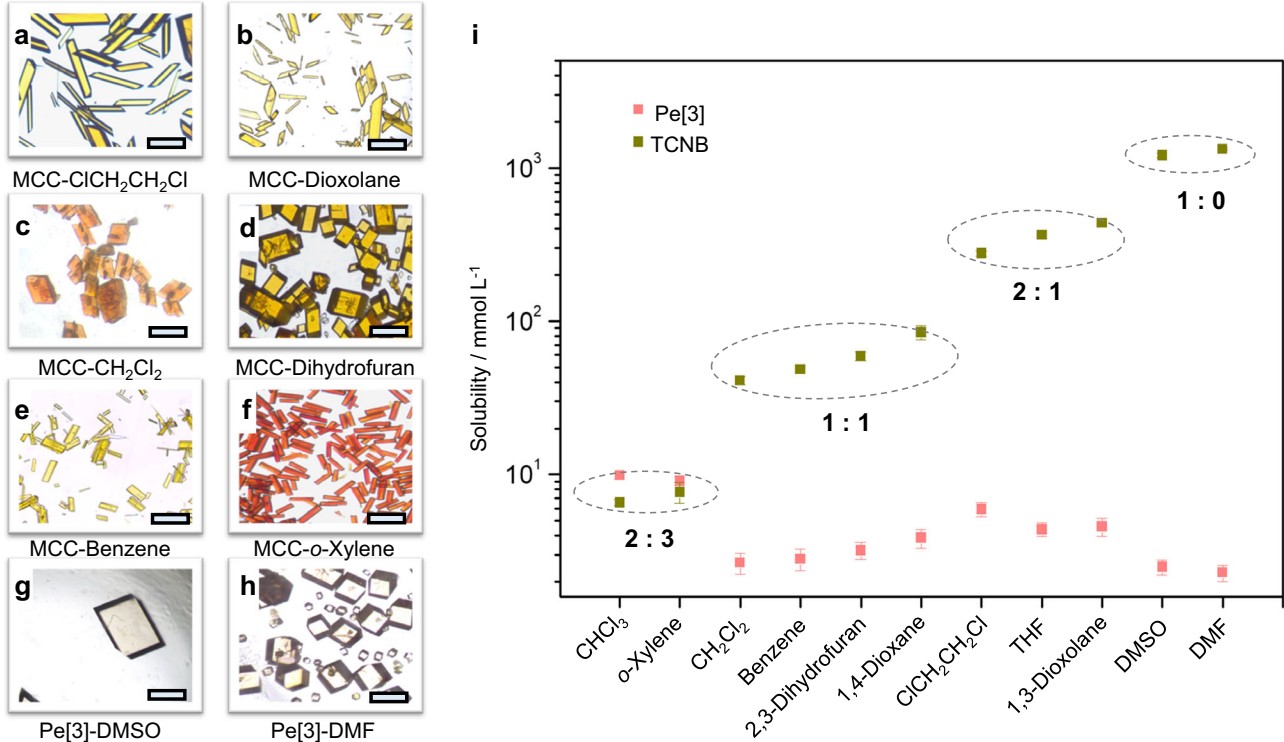

**Fig. 5 | Optical microscopy images and solubilities.** The photographs of MCCs (**a**–**f**) and individual Pe[3] crystals (**g**, **h**) in different solvents, showing various morphologies. Scale bar: 40 μm. **i** Solubility of Pe[3] and TCNB in 11 solvents at 20 °C (mean ± SD, $n = 3$).

solubility of TCNB and solvophobic forces. Besides, the photophysical properties of the 6 MCCs were carried out (Supplementary Figs. 45–58).

## Discussion

In summary, we designed and constructed three sets of MCCs with diverse D-A stoichiometric ratios. The self-assembled superstructures were constructed using an electron-rich Pe[3] macrocycle and an electron-deficient TCNB unit. Solvent modulation plays a critical role in controlling the stoichiometry and molecular arrangement of the crystal superstructures. The three MCCs exhibited tunable luminescence properties from yellow to orange, to red. The crystal structure analyses and DFT calculations revealed the

structure–property relationships. The results indicated that the CT interaction between Pe[3] and TCNB is the dominant factor of the tunable photophysical properties of these co-crystals. The results reported herein offer a deep insight into the structure–CT interaction-based luminescence relationship of the molecules and present a platform for the facile synthesis of solid-state multicolor CT luminescent materials. The fabrication of MCCs based on other functional macrocycles[40] with varying structural and functional properties exploiting CT, hydrogen bond, and halogen bond interactions is now in progress.

## Methods

### Materials

TCNB (98% purity) was purchased from the commercial source without further purification. The preparation of Pe[3][41] was provided in Supplementary Information. Nuclear magnetic resonance (NMR) spectra were recorded on Bruker Avance III 500 MHz. Chemical shifts are reported in ppm relative to the signals corresponding to the residual non-deuterated solvents ($CDCl_3$: $\delta_H = 7.26$ ppm and $\delta_C = 77.0$ ppm).

### Single crystal X-ray crystallography

All single crystal X-ray diffraction data were collected on a Bruker D8-Venture or Bruker APEX-II CCD detector using Mo-Kα ($\lambda = 0.71073$ Å) or Ga-Kα ($\lambda = 1.34138$ Å) radiation. The crystal structure was solved and refined against all $F^2$ values using the SHELX and Olex 2 suite of programs[51,52]. The detailed experimental parameters are summarized in Supplementary Tables 1–6 and 8–15.

### Fluorescence microscopy

Fluorescence microscopy images were collected at room temperature on a SOPTOP ICX41 imaging system excited at 375 nm.

### Photophysical characterization

Solid fluorescence spectra were performed with an Agilent Cary Eclipse spectrophotometer. Solid fluorescence quantum yields were performed on HAMAMATSU C9920-02 by absolute method. Fluorescence lifetimes were measured on FLS1000.

### Theoretical and computational method

The geometries of all the ground state co-crystals were selected from the corresponding X-ray single crystal diffraction data. All calculations were performed using the Gaussian 16 software package[53]. All the atomic coordinates dataset of optimized computational models are shown in Supplementary Data 1. The HOMO/LUMOs of MCCs were calculated with the popular functional B3LYP/6−31 G(d, p). The excited state property of Pe-TCNB was calculated at CAM-B3LYP/6−311 G* level. Natural transition orbitals (NTOs) were evaluated with the dominant particle-hole pair contributions and the associated transition weights. The orbitals were visualized by VMD program (the isovalue was set as 0.02) assisted with the Multiwfn program[54].

### Solubility measurements

The solubility measurements of Pe[3] and TCNB in different solvents were carried out by the gravimetric method. The excess solids are added to the desired solvent and equilibrated with constant agitation at 20 °C for 24 h. After reaching the equilibrium, the agitation was stopped for 10 min to allow the residual solid to settle. Then, taking 1 mL upper clear solution were filtered by a syringe into a preweighed glass vial. The solvent was allowed to evaporate in a fume hood at close boiling point of relative solvent for 12 h. The weight of the vial is determined. The solubility of the solids was calculated from the difference in weight.

## Data availability

The X-ray crystallographic coordinates for structures reported in this study have been deposited at the Cambridge Crystallographic Data Centre (CCDC), under deposition numbers 2159214, 2159210, 2159192, 2235489, 2285147, 2285121, 2284505, 2284851, 2284533, 2284548, 2284568, 2284821, 2285102 and 2284673. These data can be obtained free of charge from The Cambridge Crystallographic Data Centre via www.ccdc.cam.ac.uk/data_request/cif. The coordinates of the optimized structures are present as Supplementary Data 1. The authors declare that the data supporting the findings of this study are available within the paper and its Supplementary Information. All data are available from the authors upon request. Source data are provided with this paper.

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

## Acknowledgements

The authors gratefully acknowledge the National Natural Science Foundation of China (22201213, B.L. and 21971192, C.Li.), the Natural Science Foundation of Tianjin City (22JCQNJC00730, B.L. and 20JCZDJC00200, C.Li.), and the Tianjin Municipal Education Commission (2021KJ188, C.Li.).

## Author contributions

C.L. conceived the project. B.L. performed most of the experiments and wrote the manuscript; L.L. (Lingling Liu) synthesized the macrocycle compound; L.L.(Lingling Liu) and Y.W. contributed the photophysical characterizations; K.L. and Z.Z. contributed theoretical calculations; S.S., Y. H. and L.L.(Liqiang Li) carried out the OFET performances of MCCs; C.L. and B.L. finished the preparation of final manuscript; All authors discussed and commented on the paper.

## Competing interests

The authors declare no competing interests.
