## [Peer Review File · Nature Communications]

Structurally Diverse Macrocyclic Co-Crystals for Solid-State Luminescence ModulationREVIEWER COMMENTS

Reviewer #1 (Remarks to the Author):

Organic co-crystals offer an opportunity for the fabrication of advanced materials. However, conventional co-crystals are constructed by small planar molecules. They are generally packed in two stacking modes of segregated- or mixed-stacking, causing the lack of structural and functional diversity. This manuscript described interesting macrocycle-based co-crystals (MCCs) with identical co-components by combining polygonal macrocycle and co-crystal engineering. Three sets of MCCs were constructed using triangular pyrene-macrocycle (Pe[3]) and TCNB as co-formers by exo-wall CT interactions, which showed variable D-A stoichiometries (2:1, 1:1, and 2:3), diverse molecular packings and tunable emission properties. This wasn't easy to realize for conventional small-molecule co-crystals with identical co-components. Hence, I strongly recommend publishing this work in Nature Communications, pending the following minor issues being addressed.

1. The solid-state structures of MCCs seem to be solvent-rich. Although the solvents may be disordered in the structures, the authors should point out “the solvents are omitted for the sake of clarity” in the Figure legends.
2. The authors described that the monomer co-crystals of Pe-TCNB were only obtained in dioxane; please explain why the co-crystals of Pe-TCNB were not gained in THF or CHCl₃ solvents.
3. In the crystal structure of individual Pe[3], whether there are $\pi\cdots\pi$ interactions between pyrene skeletal units.
4. Some grammatical errors and formatting issues should be fixed, as detailed below.
Page 1, line 9, delete “of”.
Page 1, line 10, change “modes” to “mode”
Page 2, line 32; Page 9, line 154, change “components” to “component”
Page 7, line 118, change “superstructures” to “superstructure”.
Page 9, line 153; Page 9, line 161, change “peaks” to “peak”.
Page 13, line 219, change “Tables 1–3” to “Tables 1–4”
5. In Supplementary Figure 12, the represented “Pe[3]-TCNB-1, Pe[3]-TCNB-2, Pe[3]-TCNB-3” in the pictures should be “MCC-1, MCC-2, MCC-3”.

Reviewer #2 (Remarks to the Author):

In the manuscript titled *Structurally Diverse Macrocycle Co-Crystals for Solid-State Luminescence Modulation*, the authors reported a set of cocrystals with macrocycle Pe[3] donor and small molecule TCNB acceptor. By cocrystallization in the different solvent, the stoichiometry of D-A and molecular arrangement can be adjusted and the resulting cocrystals show different adsorption and photoluminescence properties. In general, the idea of introducing a flexible macrocycle donor with multiple interaction sites to the cocrystal for finetuning its properties is straightforward and interesting

and the superstructures of cocrystals are analyzed comprehensively. However, the manuscript seems to focus more on reporting the experimental and calculation results rather than the scientific problem beneath. Therefore, the paper is not recommended for publication in Nature Communication. There are a few issues that might need consideration:

1. Various final cocrystals could be obtained by adjusting the solvent during the cocrystallization and the stoichiometric ratios. These results are fascinating, and the mechanism beneath this solvent modulation is unfortunately missing. This is quite important for this manuscript and definitely worth to further study. Could the solvent modulation effect be due to the competition between the solvent molecules and the acceptors?
2. How is the purity of the cocrystals? The conformations of Pe[3] in MCCs are very similar. So will the formation of one crystal (for example MCCs 1) form restrict the formation of another crystal form (for example MCCs 2 or MCCs 3)? If it will, what is the mechanism? If it won't, how to make sure that it won't be disorder TCNB scattered in the cocrystals? Considering the redundancy of TCNB and the XRD only presents order structures and there might be disorder impurity.
3. From the crystal structures of MCCs1 to MCCs 3, the D-A pairs (pyrene in Pe[3] and TCNB) increased; thus the CT interactions within these cocrystals are also enhanced. Therefore, the red-shifted absorption and luminescence as well as the prolonged τ and narrowed HOMO-LUMO gap, are actually predictable as results of enhanced CT. The CT interaction is the dominant factor of the photophysical properties of these cocrystals. Thus, the insight into the structure-CT degree-based luminescence relationship claimed by the authors is not particularly accurate.
4. The frontier molecular orbitals of MCCs in Fig. 3 do not seem enough to explain the CT interaction. The HOMOs and LUMOs are concentrated on the distant pyrene moieties and TCNB, instead of the adjacent ones. In a strong CT system, the HOMO and LUMO orbitals might be just one of the many degenerate orbitals. Therefore, natural transition orbitals (NTO) analysis is recommended to give a more comprehensive analysis of the excitation process of the CT cocrystal systems.
5. The PLQY of MCC-2 is about 2-fold higher than the other two. What is the essence?
6. The average lifetimes of the three co-crystals increased from MCC-1 (7.38 ns) to MCC-2 (9.38 ns) and MCC-3 (12.66 ns), reflecting the gradual enhancement of the degree of CT realized between the donor and acceptor units. What is the theoretical basis for the increased lifetime reflecting the enhanced charge transfer?
7. Why the calculated energy gaps (2.51 eV, 2.30 eV, 1.98 eV) are not consistent with the experimental absorption wavelength of the cocrystals.
8. Several papers about different molecular packing modes obtained by identical co-constitutions (TCNB) should be cited. For example, *J. Phys. Chem. C* 2014, 118, 24688–24696; *Cryst. Growth Des.* 2014, 14, 12, 6376-6382; *ACS Appl. Mater. Interfaces* 2017, 9, 1183-1188.
9. Several papers about pyrene and TCNB cocrystal should be cited. For example, *J. Phys. Chem. C* 2021, 125, 46, 25462–25469, *J. Chem. Soc., Perkin Trans.* 1973, 2, 523-527.

Reviewer #3 (Remarks to the Author):

In this manuscript, the authors reported that they controlled stoichiometry and molecular arrangement by changing solvents and achieved tunable luminescence color of cocrystals. The authors explained the differences of luminescent colors by structural analysis and DFT calculation. Nevertheless, this work is not in high importance and novelty, and there are some issues in the work that I cannot recommend the publication on Nature Communications. More comments,

1. The phenomenon that the luminescent color of cocrystals is controlled by intermolecular charge transfer interaction is not novel or important enough. The optical properties of the cocrystals are poor, and the PLQY is significantly lower than that of monomer Pe.
2. The authors explained the intermolecular forces, but did not explain the mechanism of cocrystals formation at different stoichiometry.
3. In addition to tuning the luminescent color of the material, have these three MCCs cocrystal crystals other interesting optoelectronic applications?
4. Line 37-38, The author believed "The two stackedfunctional diversity.", which may lack rationality.
5. Line 95-96, The author described "The remarkable differencemodes for Pe[3] and TCNB (Fig. 1)." Can the degree of charge transfer be quantified? What is the relationship between the degree of charge transfer and the packing structure of the three MCCs cocrystals?
6. The authors only calculated the HOMO-LUMO band gap of three MCCs cocrystals through DFT calculations and did not effectively elaborate the relationship between charge transfer degree, packing structure, and photophysical properties.

Point-by-Point Response

Thank the reviewers for the valuable comments and suggestions on our manuscript “**Structurally Diverse Macrocycle Co-Crystals for Solid-State Luminescence Modulation**” (NCOMMS-23-06196), which are very helpful for us to improve our manuscript.

According to the comments and suggestions, the manuscript and Supplementary Information have been revised, and the point-to-point responses are as follows (for your convenience, we repeat the referee’s comments below in black, followed by our reply in blue):

Reply to Reviewer 1

Reviewer #1 (Remarks to the Author):

Organic co-crystals offer an opportunity for the fabrication of advanced materials. However, conventional co-crystals are constructed by small planar molecules. They are generally packed in two stacking modes of segregated- or mixed-stacking, causing the lack of structural and functional diversity. This manuscript described interesting macrocycle-based co-crystals (MCCs) with identical co-components by combining polygonal macrocycle and co-crystal engineering. Three sets of MCCs were constructed using triangular pyrene-macrocycle (Pe[3]) and TCNB as co-formers by exo-wall CT interactions, which showed variable D-A stoichiometries (2:1, 1:1, and 2:3), diverse molecular packings and tunable emission properties. This wasn't easy to realize for conventional small-molecule co-crystals with identical co-components. Hence, I strongly recommend publishing this work in Nature Communications, pending the following minor issues being addressed.

Reply: We sincerely appreciate the reviewer’s recommendation. According to your advice, we have made corresponding revisions and provided responses below to improve the manuscript.

1. The solid-state structures of MCCs seem to be solvent-rich. Although the solvents may be disordered in the structures, the authors should point out “the solvents are omitted for the sake of clarity” in the Figure legends.

Reply: According to your suggestions, the statement has been added in the Figure legends of the

revised manuscript and Supplementary Information.

2. The authors described that the monomer co-crystals of Pe-TCNB were only obtained in dioxane; please explain why the co-crystals of Pe-TCNB were not gained in THF or CHCl_3 solvents.

Reply: We have made many attempts to grow co-crystals of Pe-TCNB in THF or CHCl_3 , but only the crystals of the individual Pe could be obtained by slow evaporation of solvents. The single-crystal structures of Pe were shown in Figure R1, which exhibited the different molecular stacking in the two solvents. The crystallographic data were also added in the revised Supplementary Information (Supplementary Table 5 and 6).

Figure R1 (Supplementary Figure 26). The obtained solid-state packing structure of Pe when mix Pe[3] and TCNB in (a) THF and (b) CHCl_3 , which showed the different molecular stacking in the two solvents. Hydrogen atoms are omitted for clarity.

3. In the crystal structure of individual Pe[3], whether there are $\pi\cdots\pi$ interactions between pyrene skeletal units.

Reply: As can be seen Supplementary Figures 3–5, no $\pi\cdots\pi$ interactions between pyrene skeletal units were observed in the two pairs of enantiomers of individual Pe[3]. However, to the new solid-state structure of Pe[3] obtained in DMSO, offset $\pi\cdots\pi$ interactions with plane–plane distance of 3.3 Å between pyrene skeletal units were observed (Figure R2).

Figure R2. Solid-state structure of Pe[3] obtained in DMSO, showing $\pi\cdots\pi$ interactions between pyrene skeletal units.

4. Some grammatical errors and formatting issues should be fixed, as detailed below.

Page 1, line 9, delete “of”.

Page 1, line 10, change “modes” to “mode”

Page 2, line 32; Page 9, line 154, change “components” to “component”

Page 7, line 118, change “superstructures” to “superstructure”.

Page 9, line 153; Page 9, line 161, change “peaks” to “peak”.

Page 13, line 219, change “Tables 1–3” to “Tables 1–4”

Reply: Thank the reviewer very much for the suggestions, which are very helpful for improving our manuscript. According to the reviewer’s suggestions, these mistakes have been revised in the manuscript.

5. In Supplementary Figure 12, the represented “Pe[3]-TCNB-1, Pe[3]-TCNB-2, Pe[3]-TCNB-3” in the pictures should be “MCC-1, MCC-2, MCC-3”.

Reply: Done.

Reply to Reviewer 2

Reviewer #2 (Remarks to the Author):

In the manuscript titled *Structurally Diverse Macrocycle Co-Crystals for Solid-State Luminescence Modulation*, the authors reported a set of cocrystals with macrocycle Pe[3] donor and small molecule TCNB acceptor. By cocrystallization in the different solvent, the stoichiometry of D-A and molecular arrangement can be adjusted and the resulting cocrystals show different adsorption and photoluminescence properties. In general, the idea of introducing a flexible macrocycle donor with multiple interaction sites to the cocrystal for finetuning its properties is straightforward and interesting and the superstructures of cocrystals are analyzed comprehensively. However, the manuscript seems to focus more on reporting the experimental and calculation results rather than the scientific problem beneath. Therefore, the paper is not recommended for publication in Nature Communication. There are a few issues that might need consideration:

Reply: Thank the reviewer's professional comments, which are very helpful for improving our manuscript. Herewith, we addressed the referee's comments point-by-point as follows:

(1) Various final cocrystals could be obtained by adjusting the solvent during the cocrystallization and the stoichiometric ratios. These results are fascinating, and the mechanism beneath this solvent modulation is unfortunately missing. This is quite important for this manuscript and definitely worth to further study. Could the solvent modulation effect be due to the competition between the solvent molecules and the acceptors?

Reply: To explore the effect of solvents on the structure and stoichiometry of MCCs in detail, we have made many attempts. Besides the previous three co-crystals of MCC-1, MCC-2 and MCC-3, we obtained 8 new MCCs with different D-A stoichiometric ratios (2:1, 1:1, 2:3 and 1:0). The solubilities of Pe[3] and TCNB have a decisive influence on MCC formation. The solubility data were added in the revised Supplementary Information as Supplementary Table 7. The crystallographic data were also added in the revised Supplementary Information as Supplementary Tables 8–15. The related experimental method was added in the Methods section of the revised manuscript. Figure R3 was added in the revised manuscript as Figure 4; Figure R4 was added in the revised Supplementary Information as Supplementary Figure 44.

The following discussion was added to the revised manuscript in Page 12:

“Effect of solvents on MCCs formation. To investigate the effect of solvents on the structure and stoichiometry of MCCs, we made many attempts and obtained 8 sets of (co-)crystals (Supplementary Tables 8–15) from solvents with different morphologies and colors (Fig. 4a-h). Their crystal structures show that the formed D-A ratios of MCCs are 2:1 (in $\text{ClCH}_2\text{CH}_2\text{Cl}$ and 1,3-dioxolane), 1:1 (in CH_2Cl_2 , benzene and 2,3-dihydrofuran), 2:3 (in *o*-xylene) and 1:0 (in DMSO and DMF) (Supplementary Fig. 44). The varied stoichiometries of MCCs are depended on the solubility of TCNB (Fig. 4i). The lower the solubility, the stronger the solvophobic forces,⁵¹ and the CT participation ratio of TCNB is higher. Low solubility (<10 mM) of TCNB forms 2:3 MCC. Moderate solubility (41–85 mM) of TCNB tends to generate 1:1 MCC. High solubility (277–436 mM) of TCNB prefer to obtain 2:1 MCC. While TCNB solubility is more than 1000 mM, only individual macrocycles of Pe[3] were crystallized. The solubility (2.3–10 mM) of Pe[3] did not change too much in these solvents. Therefore, the effect of solvents on the diverse structures and tunable stoichiometric ratios of MCCs is decided by the solubility of TCNB and solvophobic forces.”

Figure R3 (Figure 4) Optical microscopy images and solubilities. The photographs of MCCs (a–f) and individual Pe[3] crystals (g, h) in different solvents, showing various morphologies. Scale bar: 40 μm . (i) Solubility of Pe[3] and TCNB in 11 solvents at 20 °C.

Solvents	Asymmetric Unit	Repeat Unit	Packing Structure	Stoichiometric Ratios
$\text{ClCH}_2\text{CH}_2\text{Cl}$				2 : 1
1,3-Dioxolane				2 : 1
CH_2Cl_2				1 : 1
Benzene				1 : 1
2,3-Dihydrofuran				1 : 1
o-Xylene				2 : 3
DMSO				1 : 0
DMF				1 : 0

Figure R4 (Supplementary Figure 44). The structures of MCCs with different D-A stoichiometric ratios (2:1, 1:1, 2:3 and 1:0) crystallized in different solvents. The results reveal that each packing structure differentiates from others, indicating the structural diversity of MCCs.

(2) How is the purity of the cocrystals? The conformations of Pe[3] in MCCs are very similar. So will the formation of one crystal (for example MCCs 1) form restrict the formation of another crystal form (for example MCCs 2 or MCCs 3)? If it will, what is the mechanism? If it won't, how to make sure that it won't be disorder TCNB scattered in the cocrystals? Considering the redundancy of TCNB and the XRD only presents order structures and there might be disorder impurity.

Reply: Many thanks for the valuable question. Actually, in the growth process of co-crystals, only one type of crystals was observed to precipitate at the bottom. As shown in Figure R5, the optical micrographs indicated only one homogeneous MCC with specific morphology and color was obtained. ¹H NMR experiments of MCCs were further given a strict stoichiometric ratio of MCC-1 for 2:1, MCC-2 for 1:1 and MCC-3 for 2:3 (Figure R6–R8). These results illustrate one solvent yield one co-crystal. The reasons are as follows: to MCC-3, benefitting from the similar and low solubility of cofomers and strong D-A intermolecular interactions, binary co-crystals will be precipitated first at the bottom without the single component impurity. To MCC-1 and MCC-2, due to the tremendous solubilities of TCNB in THF and dioxane, the redundancy of TCNB remained in mother liquor when the MCC precipitate. Figure R5 was added in the revised Supplementary Information as Supplementary Figure 9; Figures R6–R8 were added in the revised Supplementary Information as Supplementary Figures 10–12.

The following discussion was added to the revised manuscript in Page 5:

“The optical micrographs showed only one homogeneous crystals with specific morphology and color was formed for each MCC (Supplementary Fig. 9). ¹H NMR experiments of three MCCs were further given a strict stoichiometric ratio of MCC-1 for 2:1, MCC-2 for 1:1 and MCC-3 for 2:3, indicating the purity of MCCs (Supplementary Figs. 10–12).”

Figure R5 (Supplementary Figure 9). Optical microscopy images of (a) MCC-1, (b) MCC-2 and (c) MCC-3. Scale bar: 40 μ m.

Figure R6 (Supplementary Figure 10). ^1H NMR spectrum (400 MHz, CDCl_3 , 298 K) of MCC-1, indicating a strict stoichiometric ratio at 2:1.

Figure R7 (Supplementary Figure 11). ^1H NMR spectrum (400 MHz, CDCl_3 , 298 K) of MCC-2, indicating a strict stoichiometric ratio at 1:1.

Figure R8 (Supplementary Figure 12). ^1H NMR spectrum (400 MHz, CDCl_3 , 298 K) of MCC-3, indicating a strict stoichiometric ratio at 2:3.

(3) From the crystal structures of MCCs1 to MCCs 3, the D-A pairs (pyrene in Pe[3] and TCNB) increased; thus the CT interactions within these cocrystals are also enhanced. Therefore, the red-shifted absorption and luminescence as well as the prolonged τ_{av} and narrowed HOMO-LUMO gap, are actually predictable as results of enhanced CT. The CT interaction is the dominant factor of the photophysical properties of these cocrystals. Thus, the insight into the structure–CT degree-based luminescence relationship claimed by the authors is not particularly accurate.

Reply: Many thanks for the valuable comments. According to the reviewer's suggestion, we revised the description in the discussion section as following:

"The results indicated that the CT interaction between Pe[3] and TCNB is the dominant factor of the tunable photophysical properties of these co-crystals. The results reported herein offer a deep insight into the structure–CT interaction-based luminescence relationship of the molecules and present a platform for the facile synthesis of new solid-state multicolor CT luminescent materials."

(4) The frontier molecular orbitals of MCCs in Fig. 3 do not seem enough to explain the CT interaction. The HOMOs and LUMOs are concentrated on the distant pyrene moieties and TCNB, instead of the adjacent ones. In a strong CT system, the HOMO and LUMO orbitals might be just one of the many degenerate orbitals. Therefore, natural transition orbitals (NTO) analysis is recommended to give a more comprehensive analysis of the excitation process of the CT cocrystal systems.

Reply: To illustrate the CT interactions in the excited state, the DFT and TDDFT calculation were carried out at CAM-B3LYP/6-311G* level using the Gaussian 16 suite. We firstly attempted to calculate the natural transition orbitals (NTO) of MCCs, but we did not achieve the satisfactory results. This is possibly due to that the structures of MCCs are solvent-rich and are too complex and difficult to calculate the excitation process. Therefore, the co-crystal of the simplest Pe-TCNB structure obtained from XRD analysis was selected as the calculation model. As shown in Figure R9, the holes were delocalized over TCNB, and the particles were localized over pyrene moiety on Pe, implying CT emission. From a quantitative perspective, the model is in good agreement with the experimental results (Figure R9). Thus, we use it for qualitative analysis of CT process in MCCs. Figure R9 was added in the revised Supplementary Information as Supplementary Figure 42; Figure R10 was added in the revised Supplementary Information as Supplementary Figure 43.

The following discussion was added to the revised Supplementary Information in Page S29:

“To characterize the excited state of the co-crystals, the electronic transition densities were also calculated by the natural transition orbitals (NTOs) analysis. We selected the co-crystal structure of Pe-TCNB instead of MCCs as the calculation model because the structures of MCCs are solvent-rich and complex, which is too difficult to calculate the excitation process. As shown in Supplementary Figure 42, the holes were delocalized over TCNB, and the particles were localized over pyrene moiety on Pe, implying CT emission. From a quantitative perspective, the model is in good agreement with the experimental results (Supplementary Figure 43).”

Figure R9 (Supplementary Figure 42). The dominant natural transition orbital pairs for the first excited singlet state (the absorption peak is at 506 nm, and oscillator strength value is 0.0129). The “hole” is on the top, and the “particle” is on the bottom.

Figure R10 (Supplementary Figure 43). Solid-state UV-Vis absorption spectra of Pe, TCNB and Pe-TCNB, showing a CT band at 504 nm.

(5) The PLQY of MCC-2 is about 2-fold higher than the other two. What is the essence?

Reply: Many thanks. Although the absorption and emission are related to the D-A ratios of CT co-crystals, the PLQYs of MCCs do not have regularity.

(6) The average lifetimes of the three co-crystals increased from MCC-1 (7.38 ns) to MCC-2 (9.38 ns) and MCC-3 (12.66 ns), reflecting the gradual enhancement of the degree of CT realized between the donor and acceptor units. What is the theoretical basis for the increased lifetime reflecting the enhanced charge transfer?

Reply: Many thanks for the valuable question. In general, the excited molecule/excimer/exciple decay to ground state by radiative processes (fluorescence, phosphorescence or delay fluorescence) or nonradiative processes (molecular motions, quenching, energy transfer, etc.). The CT interactions in our co-crystal systems can suppress the nonradiative processes by restricting the vibration and rotation of chromophores. Thus, with the increase of TCNB ratios in three MCCs, the more suppressed nonradiative processes (K_{ic}) benefit to the increase of lifetime (τ). The relative references (Cryst. Growth Des. 2020, 20, 5203; Chem. Commun. 2017, 53, 2081; Cryst. Growth Des. 2018, 18, 6001;) were cited in the revised manuscript. See references 46, 49 and 50.

(7) Why the calculated energy gaps (2.51 eV, 2.30 eV, 1.98 eV) are not consistent with the experimental absorption wavelength of the cocrystals.

Reply: This is possible due to the structures of MCCs are solvent-rich and complex. Although having differences of the calculated energy gaps for MCCs, the tendency is in agreement with the results obtained from UV-Vis absorption spectra. We revised the description in the manuscript as follow:

“The gradual decrease in energy gaps (from 2.51 eV to 2.30 eV and subsequently to 1.98 eV) demonstrated an increased CT interaction. The tendency is in agreement with the results obtained from UV-Vis spectral profiles. The decrease in bandgap can be attributed to the change in the D-A ratio of the MCCs.”

(8) Several papers about different molecular packing modes obtained by identical co-constitutions (TCNB) should be cited. For example, J. Phys. Chem. C 2014, 118, 24688–24696; Cryst. Growth Des. 2014, 14, 12, 6376-6382; ACS Appl. Mater. Interfaces 2017, 9, 1183-1188.

Reply: The recommended references are very valuable to our work. We have referenced them in the introduction section. Please see references 21–23.

(9) Several papers about pyrene and TCNB cocrystal should be cited. For example, J. Phys. Chem. C 2021, 125, 46, 25462–25469, J. Chem. Soc., Perkin Trans. 1973, 2, 523-527.

Reply: The two references have been cited in the revised manuscript. Please see references 43 and 44.

Reply to Reviewer 3

Reviewer #3 (Remarks to the Author):

In this manuscript, the authors reported that they controlled stoichiometry and molecular arrangement by changing solvents and achieved tunable luminescence color of cocrystals. The authors explained the differences of luminescent colors by structural analysis and DFT calculation. Nevertheless, this work is not in high importance and novelty, and there are some issues in the work that I cannot recommend the publication on Nature Communications. More comments,

Reply: We greatly appreciate reviewer's comments. Herewith, we addressed the referee's comments as follows:

1. The phenomenon that the luminescent color of cocrystals is controlled by intermolecular charge transfer interaction is not novel or important enough. The optical properties of the cocrystals are poor, and the PLQY is significantly lower than that of monomer Pe.

Reply: In traditional small-molecule co-crystals, the luminescent regulation by CT interaction is common. However, our work combines the macrocyclic chemistry and co-crystal engineering to

construct novel macrocycle co-crystals (MCCs). The luminescence behavior can be tuned by controlling the D-A stoichiometric ratio in MCCs. Moreover, compared with traditional co-crystals, MCCs have some unique advantages: 1) the topological structure of polygon for macrocycles makes the supramolecular arrangements and stoichiometric ratios between components to be more diverse (In this work, we obtained 9 MCCs with three stoichiometric ratios and their molecular packing arrangements are distinct from each other); 2) the introduction of macrocycles endows organic co-crystals new recognition and efficient adsorption capacity; 3) it is more convenient to realize the construction of multi-component co-crystals by using exo-wall binding and in-cavity complexation. **In a word, our work proves that the marriage of polygonal macrocycles and co-crystal engineering provides a smart strategy to build organic co-crystals with variable stoichiometric ratio and diverse supramolecular structures, and the utilization of exo-wall interactions also provides a new perspective and opportunity for design and construction of supramolecular organic materials.** Thus, we believe that our work is significant and novelty enough to be published in *Nature Communications*.

2. The authors explained the intermolecular forces, but did not explain the mechanism of cocrystals formation at different stoichiometry.

Reply: Thank the reviewer's very useful suggestions. To explore the effect of solvents on the structure and stoichiometry of MCCs in detail, we made many attempts. Besides the previous three co-crystals of MCC-1, MCC-2 and MCC-3, 8 new MCCs with different D-A stoichiometric ratios (2:1, 1:1, 2:3 and 1:0) were obtained. The solubilities of Pe[3] and TCNB have a decisive influence on MCC formation. Please refer to "Reply to Reviewer 2" in detail.

3. In addition to tuning the luminescent color of the material, have these three MCCs cocrystal crystals other interesting optoelectronic applications?

Reply: Many thanks for the suggestion. Organic D-A co-crystals extend the semiconductor family and promote charge separation and transport in organic field-effect transistors (OFETs). Considering the varied D-A ratios and diverse structures of MCCs, we attempted to explore the OFET performances of these three MCCs and their films fabricated by a solution spin-coating method. As shown in Figure R11, all devices have extremely low source-drain current and no effective on/off ratio, indicating the charge transfer performances of these three MCCs materials

are poor. This can be attributed to the lower conjugated structure of the flexible macrocycle (rotation of pyrene skeleton). We are building the MCCs materials with high conjugated degree in our lab and will continue to explore their optoelectronic applications in the future.

Figure R11. The transfer characteristics of OFET device based on MCCs (a–c) and MCC films fabricated by a solution spin-coating method (d–f).

4. Line 37-38, The author believed “The two stackedfunctional diversity.”, which may lack rationality.

Reply: Many thanks. We deleted this sentence in the revised manuscript.

5. Line 95-96, The author described “The remarkable differencemodes for Pe[3] and TCNB (Fig. 1).” Can the degree of charge transfer be quantified? What is the relationship between the degree of charge transfer and the packing structure of the three MCCs cocrystals?

Reply: The CT participation ratio of macrocycle sidewall in MCC-1, MCC-2 and MCC-3 are 1/3, 2/3 and 3/3, respectively. The following discussion was added in the revised manuscript:

“The remarkable difference among the structures of the three co-crystals can be attributed to the CT participation ratio of macrocyclic skeleton (1/3 for MCC-1, 2/3 for MCC-2 and 3/3 for MCC-3) (Fig. 1).”

6. The authors only calculated the HOMO-LUMO band gap of three MCCs cocrystals through DFT

calculations and did not effectively elaborate the relationship between charge transfer degree, packing structure, and photophysical properties.

Reply: Besides the calculation of HOMO-LUMO band gap, we also attempted to calculate the natural transition orbitals (NTO) of MCCs at CAM-B3LYP/6-311G* level to give a more comprehensive analysis of the excitation process of the CT cocrystal systems and to elaborate the relationship between CT interaction and photophysical properties. However, because the structures of MCCs are solvent-rich and complex, we did not achieve the satisfactory results. Therefore, we selected the co-crystal of the simplest Pe-TCNB structure obtained from XRD analysis as the calculation model. From a quantitative perspective, the model is in good agreement with the experimental results (Figure R9 and R10). The detailed discussion was added in the revised Supplementary Information on Page S29.

REVIEWER COMMENTS

Reviewer #2 (Remarks to the Author):

The author answer all my concerned, I suggest to publish in Nature Communication.

Reviewer #3 (Remarks to the Author):

In this revised manuscript, the authors investigated the different solvent effects on the cocrystal ratios and packing structures. The solvophobic force modulating the co-assembly ratio of MCCs proposed in the manuscript is interesting. However, the authors did not investigate the photophysical properties of the new MCCs. They did not deeply explored the relationship between packing structures and optical properties. Therefore, we do not recommend the paper for publication in Nature Communication. More comments:

1. Some interesting studies have been reported on macrocyclic cocrystals (Angew. Chem. Int. Ed. 2022, 61, e202117872. Nat Commun 2020, 11, 4633. J. Am. Chem. Soc. 2019, 141, 17783-17795. Sci. China Mater. 2021, 64, 1510–1514.), and these structures realize novel optoelectronic properties or fascinating packing patterns. In this manuscript, the three MCCs have maximum PLQY of only 2.7%, and no relevant OFET performance has been measured, which does not seem to be novel among the reported macrocyclic cocrystal systems.
2. In reply to question 2, “.....8 new MCCs with different D-A stoichiometric ratios (2:1, 1:1, 2:3 and 1:0) were obtained.” is unreasonable. In fact, the crystals with D-A ratios of 1:0 cannot belong to cocrystals.
3. In reply to question 5, the authors believes that the degree of charge transfer is related to the sidewall participation rate. However, they did not provide reasonable evidences.

Reviewer #4 (Remarks to the Author):

In the work the authors reported a set of cocrystals with macrocycle Pe[3] donor and small molecule TCNB acceptor. By cocrystallization in the different solvent, the stoichiometry of D-A and molecular arrangement can be adjusted and the resulting cocrystals show different adsorption and photoluminescence properties. In general, the idea of introducing a flexible macrocycle donor with multiple interaction sites to the cocrystal for finetuning its properties is straightforward and interesting and the superstructures of cocrystals are analyzed comprehensively.

The authors provide comprehensive and persuasive clarification. This referee strongly recommend publishing this work in Nature Communications.

The editorial board has requested supplementary comments:

Referee 1's comments exhibit a high level of precision; however, they do not bring forth any notably significant problems or uncertainties. Instead, they seek clarification on minor issues:

1. The solid-state structures of MCCs seem to be solvent-rich. Although the solvents may be disordered in the structures, the authors should point out “the solvents are omitted for the sake of clarity” in the Figure legends.
2. The authors described that the monomer co-crystals of Pe-TCNB were only obtained in dioxane; please explain why the co-crystals of Pe-TCNB were not gained in THF or CHCl₃ solvents.
3. In the crystal structure of individual Pe[3], whether there are π ... π interactions between pyrene skeletal units.
4. Some grammatical errors and formatting issues should be fixed, as detailed below.

Contrastingly, referees 2 and 3 did not endorse the paper for publication in Nature Communications. They highlighted several concerns that may require attention if the paper is to be considered for publication in alternative journals

Point-by-Point Response

Reviewer #2 (Remarks to the Author):

The author answer all my concerned, I suggest to publish in Nature Communication.

Reply: We sincerely appreciate the reviewer's recommendation.

Reviewer #3 (Remarks to the Author):

In this revised manuscript, the authors investigated the different solvent effects on the cocrystal ratios and packing structures. The solvophobic force modulating the co-assembly ratio of MCCs proposed in the manuscript is interesting. However, the authors did not investigate the photophysical properties of the new MCCs. They did not deeply explored the relationship between packing structures and optical properties. Therefore, we do not recommend the paper for publication in Nature Communication. More comments:

Reply: Thank the reviewer's comments, which are very helpful for improving our manuscript. According to the reviewer's suggestions, we measured the photophysical properties (solid-state ultraviolet-visible (UV-vis) absorption and fluorescence spectra, photoluminescence quantum yield and fluorescence lifetime) of the 6 new MCCs. The relative data were shown in Figure R1–R14 (Supplementary Figures 45–58).

The following discussion was added to the revised Supplementary Information in Page 38:

"As shown in Supplementary Figures 45 and 46, the solid-state ultraviolet-visible (UV-vis) absorption and fluorescence spectra shown that the MCCs significantly red-shifted with the increase of CT participation ratio of macrocyclic skeleton (from 2:1 to 1:1 to 2:3). The solid-state PLQYs of MCC-ClCH₂CH₂Cl, MCC-Dioxolane, MCC-CH₂Cl₂, MCC-Dihydrofuran, MCC-Benzene and MCC-o-Xylene were 0.4, 1.4, 1.2, 1.5, 5.1 and 2.7%, respectively (Supplementary Figures 47–52). The average lifetimes of the 6 MCCs displayed an increasing tendency with the increase of CT participation ratio of macrocyclic skeleton (Supplementary Figures 53–58)."

Figure R1 (Supplementary Figure 45). Solid-state UV-Vis absorption spectra of MCC-ClCH₂CH₂Cl (475 nm), MCC-Dioxolane (488 nm), MCC-CH₂Cl₂ (490 nm), MCC-Dihydrofuran (496 nm), MCC-Benzene (505 nm) and MCC-*o*-Xylene (537 nm).

Figure R2 (Supplementary Figure 46). Solid-state fluorescence spectra of MCC-ClCH₂CH₂Cl (E_m : 527 nm), MCC-Dioxolane (E_m : 570 nm), MCC-CH₂Cl₂ (E_m : 605 nm), MCC-Dihydrofuran (E_m : 602 nm), MCC-Benzene (E_m : 613 nm) and MCC-*o*-Xylene (E_m : 637 nm).

Figure R3 (Supplementary Figure 47). Quantum yield of MCC-ClCH₂CH₂Cl.

Figure R4 (Supplementary Figure 48). Quantum yield of MCC-Dioxolane.

Figure R5 (Supplementary Figure 49). Quantum yield of MCC-CH₂Cl₂.

Figure R6 (Supplementary Figure 50). Quantum yield of MCC-Dihydrofuran.

Figure R7 (Supplementary Figure 51). Quantum yield of MCC-Benzene.

Figure R8 (Supplementary Figure 52). Quantum yield of MCC-o-Xylene.

Figure R9 (Supplementary Figure 53). Time-resolved fluorescence decay curve of MCC-ClCH₂CH₂Cl at 527 nm in the solid state.

Figure R10 (Supplementary Figure 54). Time-resolved fluorescence decay curve of MCC-Dioxolane at 570 nm in the solid state.

Figure R11 (Supplementary Figure 55). Time-resolved fluorescence decay curve of MCC-CH₂Cl₂ at 605 nm in the solid state.

Figure R12 (Supplementary Figure 56). Time-resolved fluorescence decay curve of MCC-Dihydrofuran at 602 nm in the solid state.

Figure R13 (Supplementary Figure 57). Time-resolved fluorescence decay curve of MCC-Benzene at 613 nm in the solid state.

Figure R14 (Supplementary Figure 58). Time-resolved fluorescence decay curve of MCC-o-Xylene at 637 nm in the solid state.

1. Some interesting studies have been reported on macrocyclic cocrystals (Angew. Chem. Int. Ed. 2022, 61, e202117872. Nat Commun 2020, 11, 4633. J. Am. Chem. Soc. 2019, 141, 17783-17795. Sci. China Mater. 2021, 64, 1510–1514.), and these structures realize novel optoelectronic properties or fascinating packing patterns. In this manuscript, the three MCCs have maximum PLQY of only 2.7%, and no relevant OFET performance has been measured, which does not seem to be novel among the reported macrocyclic cocrystal systems.

Reply: Compared with the reported macrocyclic cocrystal systems, our synthesized macrocycle donor has more flexible scaffold bearing various molecular configurations, which can readily tune the stoichiometry ratios and stacking structure of co-crystals. In this work, we obtained 9 sets of MCCs with different stoichiometric ratios (2:1, 1:1 and 2:3) and packing structures by regulating solvent. Moreover, we revealed the effect of solvents on the diverse structures and tunable stoichiometric ratios of MCCs is decided by the solubility of TCNB and solvophobic forces. Although these MCCs have lower PLQY, this work provides novel strategy and guidance to construct structurally diverse MCCs. In the future work, we will attempt to get more MCCs to realize novel optoelectronic properties.

2. In reply to question 2, “.....8 new MCCs with different D-A stoichiometric ratios (2:1, 1:1, 2:3 and 1:0) were obtained.” is unreasonable. In fact, the crystals with D-A ratios of 1:0 cannot belong to cocrystals.

Reply: Thank you for pointing out the mistake. In fact, we have described “we made many attempts and obtained 8 sets of (co-)crystals.....with different morphologies and colors” in the initial revised manuscript. To avoid misunderstanding, we revised “8 sets of (co-)crystals” to “6 sets of MCCs and 2 individual macrocycle crystals” in the revised manuscript.

The following discussion was added to the revised manuscript in Page 11:

“To investigate the effect of solvents on the structure and stoichiometry of MCCs, we made many attempts and obtained 6 sets of MCCs and 2 individual macrocycle crystals (Supplementary Tables 8–15) from solvents with different morphologies and colors (Fig. 4a-h).”

3. In reply to question 5, the authors believes that the degree of charge transfer is related to the sidewall participation rate. However, they did not provide reasonable evidences.

Reply: In the revised manuscript, we did not mention “the degree of charge transfer” but “the CT participation ratio of macrocyclic skeleton”. The CT participation ratio can be clearly proved through the single crystal structures of MCCs.

Reviewer #4 (Remarks to the Author):

In the work the authors reported a set of cocrystals with macrocycle Pe[3] donor and small molecule TCNB acceptor. By cocrystallization in the different solvent, the stoichiometry of D-A and molecular arrangement can be adjusted and the resulting cocrystals show different adsorption and photoluminescence properties. In general, the idea of introducing a flexible macrocycle donor with multiple interaction sites to the cocrystal for finetuning its properties is straightforward and interesting and the superstructures of cocrystals are analyzed comprehensively.

The authors provide comprehensive and persuasive clarification. This referee strongly recommend publishing this work in Nature Communications.

Reply: We sincerely appreciate the reviewer's recommendation.

REVIEWERS' COMMENTS

Reviewer #3 (Remarks to the Author):

The authors have carefully addressed my questions and comments, I would like to recommend the revised version for publication in Nature Communication.

REVIEWER COMMENTS

Reviewer #3 (Remarks to the Author):

The authors have carefully addressed my questions and comments, I would like to recommend the revised version for publication in Nature Communication.

Reply: We sincerely appreciate the reviewer's recommendation.